# Can Artificial Intelligence Help in the Study of Vegetative Growth Patterns from Herbarium Collections? An Evaluation of the Tropical Flora of the French Guiana Forest

**DOI:** 10.3390/plants11040530

**Published:** 2022-02-16

**Authors:** Hervé Goëau, Titouan Lorieul, Patrick Heuret, Alexis Joly, Pierre Bonnet

**Affiliations:** 1Botany and Modeling of Plant Architecture and Vegetation (AMAP), French Agricultural Research Centre for International Development (CIRAD), French National Centre for Scientific Research (CNRS), French National Institute for Agriculture, Food and Environment (INRAE), Research Institute for Development (IRD), University of Montpellier, 34398 Montpellier, France; patrick.heuret@inrae.fr (P.H.); pierre.bonnet@cirad.fr (P.B.); 2ZENITH Team, Laboratory of Informatics, Robotics and Microelectronics-Joint Research Unit, Institut National de Recherche en Informatique et en Automatique (INRIA) Sophia-Antipolis, CEDEX 5, 34095 Montpellier, France; titouan.lorieul@inria.fr (T.L.); alexis.joly@inria.fr (A.J.)

**Keywords:** vegetative phenology, plant growth, herbarium specimens, natural history collection, deep learning, machine learning, neural network, visual analysis

## Abstract

A better knowledge of tree vegetative growth phenology and its relationship to environmental variables is crucial to understanding forest growth dynamics and how climate change may affect it. Less studied than reproductive structures, vegetative growth phenology focuses primarily on the analysis of growing shoots, from buds to leaf fall. In temperate regions, low winter temperatures impose a cessation of vegetative growth shoots and lead to a well-known annual growth cycle pattern for most species. The humid tropics, on the other hand, have less seasonality and contain many more tree species, leading to a diversity of patterns that is still poorly known and understood. The work in this study aims to advance knowledge in this area, focusing specifically on herbarium scans, as herbariums offer the promise of tracking phenology over long periods of time. However, such a study requires a large number of shoots to be able to draw statistically relevant conclusions. We propose to investigate the extent to which the use of deep learning can help detect and type-classify these relatively rare vegetative structures in herbarium collections. Our results demonstrate the relevance of using herbarium data in vegetative phenology research as well as the potential of deep learning approaches for growing shoot detection.

## 1. Introduction

The current challenges of climate change require a better understanding of plant growth dynamics, particularly to better measure the evolution of the carbon sequestration capacity of forests [1]. The dynamics of carbon sequestration are intimately linked to the phenology of growth processes in woody plants, i.e., the temporal patterns of recurrent biological events [2]. A tree is an organism that builds on itself by extending its axes (stems and roots) through the process of primary growth and by thickening these axes via the formation of wood through the process of secondary growth [3]. The accumulation of carbohydrates in wood represents the first component of carbon sequestration in trees. Because carbohydrates are synthesized in leaves through photosynthesis, it is of primary importance to understand shoot extension dynamics (which include shoot leaf and internode extension), leaf functioning, and life span over time. This understanding must nevertheless be achieved by integrating different scales such as the leaf organ, the crown of the tree, the populations of species, or the community of species that make up the forest canopy.

In temperate climates, the phenology of the fundamental processes underlying tree development and reproduction is coordinated by temperature and light constraints imposed by winter [4]. Broadly speaking, the phenology of trees in northern temperate regions generally follows these successive steps: (i) bud break and appearance of the first floral structures between late winter and early spring, (ii) flowering and development of vegetative new shoots during spring, (iii) fruiting and slowing down of vegetative growth during summer, (iv) dissemination of seeds and fruits coupled with defoliation during autumn, and (v) plant dormancy during winter. An illustration of that pattern is provided in Figure 1 for four common tree species in western European temperate regions. Therefore, in those regions, an event such as growth can be said to occur, with annual periodicity, in spring. It lasts from a few days to several weeks, depending on the species, with an amplitude—e.g., the length of the shoot—that varies according to the species, the environment of the tree, or the shoot position inside the tree architecture; it is also synchronous, both at the individual tree and the species population levels. This simplified description may seem rough, and one could propose numerous exceptions (e.g., buds break more or less early in the spring depending on the individuals of a population, several shoot extensions can appear within a year due to polycyclism, etc.); however, compared to the diversity of phenological patterns observed in the tropics, it remains coherent.

The great diversity of species present in the tropics is attended by a wide variety of life forms and phenological patterns [5,6]. Except for high-altitude locations, in those tropical regions, temperature is generally considered not to be a limiting and structuring trigger in phenological growth patterns [7]. Rather, variations in insolation or water availability appear to be the main drivers at the tree community level [8,9,10]. To illustrate this point, we consider French Guiana, which will be used as a support for this study. An overall phenological pattern at the scale of the tree community can be observed, but it is not as marked as in temperate climates. Indeed, based on regular monitoring of the vegetation by different approaches (direct observations on transects, nets placed in the understory collecting twigs, leaves, flowers, fruits that fall from the canopy, and observations by remote sensing), a general pattern emerges at the global scale of the forest. In the period of August–September, at the very beginning of the long dry season (precipitation < 100 mm·month) when solar irradiation is at its maximum (minimal cloud cover and high length of daylight), the trees renew their leaves. These shoots bear flowers and the peak of flowering occurs in September–October. With the return of the rains, the fertilized flowers are transformed into fruits, which will take several months to mature, and the peak of fruiting is reached in April of the following year, in the heart of the long rainy season. In July–August, many leaves fall from the canopy and deciduous species shed their leaves [11,12,13,14]. However, behind this general pattern lies a strong inter-annual variability in the calendar of these events. More importantly, at the species level, many of them have phenological patterns deviating from this general seasonal pattern. For instance, some species can grow at different times of the year with populations whose individuals are not necessarily synchronous with each other [12,15]. Furthermore, the patterns of shoot extension over time (e.g., rhythm, duration) remain largely unknown at the species level. It is thus difficult to understand how we can change the scale, from a collection of species-specific phenological patterns to a global phenological pattern at the community scale, all in interaction with environmental climate fluctuations. Herbarium collections could provide a valuable source of data to help fill this knowledge gap.

The recent and massive digitization of natural history collections, particularly of herbaria, offers new material available for many research activities [16,17,18] such as the analysis of plant species morphology and diversity [19,20]. Because most of the collections are dated and geolocated, they constitute a valuable source of information for (i) determining the proven or potential distribution areas of species [21,22,23], whether native or exotic (dynamics of biological invasions), with direct applications in conservation biology [24], and for (ii) determining the reproductive phenological patterns of species (e.g., date and duration of flowering and fruiting periods) [25,26,27,28]. Research in these fields has been particularly stimulated by questions related to climate change and its effect on the range of species distribution [29,30] or their biological rhythms [31,32,33,34,35,36,37,38,39]. More original aspects have been addressed such as changes over time in (i) herbivory [40,41], (ii) the concentration of isotopes (δC13, δO18) related to water use efficiency or photosynthetic efficiency [42], or (iii) the diversity of endophytic fungi present in leaves [43]. This newly digitized material, widely available through platforms such as iDigBio [44] and eReColNat (https://www.recolnat.org/en/ accessed on 10 May 2021) provides access to herbarium specimen scans in addition to textual data. These images contain very rich information that, before being exploited, needs to be extracted, for instance through crowdsourcing platforms [38] or automated approaches [45,46]. Among these last approaches, recent works using deep learning technologies for the analysis of these data have made it possible to evaluate the relevance of these techniques for the identification of species [47,48], the detection of the reproductive phenological status [49], and the estimation of the number of reproductive structures [50,51,52]. Despite the diversity of these works, none of them has so far evaluated the use of these techniques for the detection of young growing shoots. This can be explained in part by the fact that plant samples are mainly collected with reproductive structures rather than with growing vegetative shoots alone. Indeed, since plant identification at the species level is largely based on the analysis of reproductive structures, a collector will generally select plant specimens with flowers or fruits rather than those solely with leaves. Such fertile specimens therefore often have a greater contribution to subsequent research studies than sterile specimens.

In this study, we propose to evaluate the potential of deep learning approaches for the analysis of the vegetative axis extension pattern of tropical tree species based on the analysis of herbarium specimens. To the best of our knowledge, no such study has ever been conducted. Hence, this work could pave the way for a new form of research on plant primary growth dynamics based on such new material. The challenges are (i) to automatically detect a young shoot bearing leaves in extension, (ii) to detect the modalities of growth expression (e.g., the presence of growth units due to rhythmic growth), and even (iii) to eventually distinguish different stages of leaf shoot maturation (phenophases). In the presence of a sufficient amount of detected herbarium samples, and by leveraging their dates of collection, this would then allow for a study of the timing and synchronicity of growth events within populations and communities by analyzing the frequencies of the detected events [27]. With regards to existing deep learning methods, several challenges must be addressed. First, since specimen collection is primarily driven by the presence of reproductive organs, one can expect to find few samples showing vegetative growing shoots, making it difficult to train an automatic detection model based on deep learning, which requires a rather large number of training examples to learn from. Moreover, due to the great taxonomic diversity of tropical regions and the numerous and various visual aspects and potential smallness of young growing shoots, the automatic detection model needs to be robust to size and capable of generalizing to a wide variety of visual patterns.

Thus, the main motivation of this work is to try to answer the following questions:Do herbaria contain relevant information for the study of tropical tree growth? If so, to what extent?Can deep learning-based automated approaches detect growing specimens?If so, which approaches are most relevant and which growth patterns are best detected?

## 2. Results

### 2.1. Herbarium Data

A new dataset was created from a selection of specimens located in different ecosystems of the French Guiana territory, scanned and hosted by the Herbarium of French Guiana (http://herbier-guyane.ird.fr/ accessed on 10 May 2021, see Section 4.1 for more details on the various steps involved in assembling and enriching the dataset). Each specimen was annotated with taxonomic and contextual information (species, genus, family, date, name of locality, and GPS), with tags on the presence of fertile organs, but initially without information on the presence or absence of vegetative growths. A great effort was made by two experts to manually and meticulously select relevant specimens for our experiments by checking for the presence or absence of new growing shoots. In the end, the resulting dataset consists of 408 herbarium specimens belonging to 57 species, 40 genera, and 25 families, with a total of 77 specimens showing the presence of recent shoots. Two complementary tags specifying the type of growing shoots—“Rhythmic” or “Continuous”, corresponding to two fundamentally different plant growth behaviors—were added by the experts to those 77 specimens. Finally, the scans of each specimen were manually annotated to provide bounding boxes and masks that precisely localize the growth shoots in the image. These were used for the training of deep model approaches evaluated in the experiments. The resulting dataset and its additional annotations of growing vegetative shoots (type of shoot, bounding boxes, and masks) are an important first contribution of this paper by providing a new annotated dataset, opening perspectives to phenological analyses based on the observation of this type of botanical structure.

### 2.2. Global Model Detection Performance (EXP1, ResNet50)

Three deep learning architectures were evaluated, which can be distinguished as *global* and *local* approaches (see Section 4.3 for more details about the models used and the design of the experiments). The global approach is based on a ResNet50 convolutional neural network working at the scale of an entire image without specifying which subparts do or do not belong to a growing shoot, as in a standard image classification task. On the other hand, the two local approaches are based on convolutional neural networks specifically designed for object detection tasks respectively exploiting two different levels of annotation: (i) bounding boxes for a Faster R-CNN model, and (ii) fine contour delineations of the growing shoots for a Mask R-CNN model. The Receiver Operating Characteristic (ROC), a classical performance measure for detection tasks that reports the True Positive Rate versus the False Positive Rate, was used for each experiment.

The performance of the global model is shown in Figure 2. For growing shoot detection, this model significantly outperforms a random classifier despite the difficulty of the task. It is thus able to capture discriminant information using only a global view of the image without any form of localization information of the growing shoot(s).

For comparison, we also tested the performance of this model for a fertility detection task on exactly the same herbarium specimens used for both training and assessment (annotations related to the presence and absence of reproductive structures are based on previous work [49]). Note that due to the selected specimens for these experiments, the ratio of fertile specimens is around 10–15%. This proportion is much lower than the fertility ratio of the rest of the specimens in CAY herbaria (around 79.4%), but comparable to the ratio of specimens possessing growing shoots (18.87%, according to Table 1). Figure 2 shows that the global approach is significantly better for fertility detection than for growing shoot detection. This highlights the fact that detecting the presence of flowers and/or fruits is much easier than that of growing shoots.

The performance of the global approach emphasizes the need for a more adapted method that takes into account the specificity of the task. In the next section, we study the performance of methods specifically designed for object detection.

### 2.3. Local Models Detection Performance (EXP2, Faster R-CNN, and Mask R-CNN)

The performance of the local approaches—Faster R-CNN using bounding boxes and Mask R-CNN using masks—measured by ROC curves are given in Figure 3 in a similar way to Figure 2. Both local approaches performed better overall than the global approach, with better trade-offs between the true positive rate (TPR) and the false positive rate (FPR). The optimal cut-off point is defined as the point closest to the upper-left corner, and it indicates the best compromise between TPR and FPR for each model. For the Faster Region based Convolutional Neural Network (Faster R-CNN) model, it is attained for a TPR of 0.80 and an FPR limited to 0.25. For the Mask R-CNN model, there is no salient point along the curve, but a cut-off point can be found for a TPR of 0.92 and an FPR of 0.46. To give a more condensed view of the results and to facilitate the comparison of the three methods, Table 2 reports the true positive rates given at three regular intervals of false negative rates (0.10, 0.20, and 0.30). As an example, this table indicates that if 20% of the images without growing shoots are incorrectly predicted as possessing growing shoots, by using the Faster R-CNN model, we can expect to find 64% of all the specimens really containing a growing shoot.

### 2.4. Local Models Two-Class Detection Performance (EXP3, Faster R-CNN, and Mask R-CNN)

In the previous experiments, during training, the models were told if a scan contained one or several growing shoots (global approach in EXP1) and then given their localization in the image (local approaches in EXP2). The additional information of the type of growing shoots—“Continuous” or “Rhythmic”—can also be provided and leveraged by the detection models. This is what was studied in experiment EXP3.

Figure 4 provides the ROC curves of the Faster R-CNN and Mask R-CNN models. This time, the Mask R-CNN model seems to have outperformed the Faster R-CNN as compared to the previous experiment, EXP2. Indeed, the Faster R-CNN model has more difficulties in correctly detecting the “Rhythmic” category with an FPR and TPR close to that of random classifiers. On the contrary, the Mask R-CNN model obtained ROC curves with high AUC (area under the curve) for both categories, and with optimal cut-off FPR and TPR values of 0.28 and 0.90, respectively, for the “Continuous” category, and optimal cut-off FPR and TPR values of 0.31 and 0.87, respectively, for the “Rhythmic” category. Similarly to EXP2, Table 3 gives a more condensed view of the performances of the models by providing the TPR values of the two local models and the two types of growing shoots at three chosen FPR values (0.10, 0.20, and 0.30). It highlights the fact that the Mask R-CNN model returns more correctly detected images than the Faster R-CNN model for both types of growing shoots if we tolerate that 20% or more of the images without growing shoots are incorrectly predicted with growing shoots.

## 3. Discussion

This study confirms the fact that the proportion of specimens showing new growing shoots is extremely low in herbarium collections. During the preparation of the dataset, the two experts went through 1109 herbarium sheets to certify that 104 pictures (9.46%) contain growing shoots. When we compare this proportion to that of flowering plants on all the specimens of the herbarium on which [49] have worked (ranging from 79.4 to 92.7%), we see a highly significant difference. This difference can be interpreted by the combined facts that (i) the elongation and maturation of new shoots usually occur over a relatively short period of time when growth is rhythmic and that (ii) botanists preferentially collect fertile (i.e., flowering or fruiting) specimens in order to have maximum evidence for their botanical determination. Because of the long period of fruit ripening, the axes are thus fertile for many months of the year. In fact, only few specimens with a new growth shoot were in flower or fruit during our study. This highlights the low concomitance of the presence of growth shoots and the reproductive structure and partly explains why we observe so few specimens with growing shoots in the studied specimens.

Our study shows that despite the small number of samples and the size and diversity of visual patterns, automatic detection technologies can nowadays partially predict the presence of growing shoots in herbarium collections. These results are promising and encourage the performance of more investigations and intensive evaluation on larger volumes of training and test data.

The performance of the three types of approaches are proportional to the annotation effort in the images: the limited annotations for the presence/absence tag of the growing shoots led to the least good performances, while a time-consuming annotation effort, with masks delimiting the growing shoots to the nearest pixel and a categorization of the type of shoot allowed for the achievement of the best results. Intermediate annotation effort with bounding boxes and a single general category of growing shoot offered an interesting compromise between annotation effort and performance.

In the case of a single general category of growing shoots, it seems that the Mask R-CNN approach encounters more difficulties to converge than the Faster R-CNN approach. This observation could be explained by the fact that the mask prediction learning was forced on more heterogeneous visual concepts, which are more difficult to generalize from and would require more data for the model to capture appropriately. Conversely, if the two concepts of growing shoots are considered as two distinct categories, this additional a priori information will be leveraged by the model. In this case, the Mask R-CNN can more easily capture the contours of both types of growing shoots, converge better, and finally outperform the Faster R-CNN.

Detecting “Rhythmic” growth is more difficult than detecting “Continuous” growth, especially with the Faster R-CNN model, and to a lesser extent with the mask R-CNN model. Shoots labeled “continuous growth” mostly develop at the ends of branches and are then often arranged on the uniform background of the sheet without overlapping with other parts of the plant. Conversely, rhythmic shoots include lateral branching and can more frequently partially cover or be covered by other parts of the plant. Bounding boxes around a shoot then capture a lot of less relevant content (parts of mature leaves, bark, tape) that probably disturbs the training of the Faster R-CNN model on the rhythmic growing shoots.

In the best of cases, when using the Mask R-CNN model, about 87–90% of the herbaria with growing shoots can be found at a rate of 28–30% false alarms according to the current evaluated test set. Furthermore, it is interesting to note that the Mask R-CNN offers the additional possibility of producing masks on which biologists could conduct morphological measurements (size, shape, coloration), bringing more fineness to such types of herbarium-based phenological analyses. However, this approach deserves more investigations on the quality of the masks produced and the risk of overfitting because it seems that predicted masks of better quality are created at the cost of more training iterations, leading to a decrease in the true positive rate (see Appendix B).

Deep learning, and AI in general, is a very active research field. The performance of such difficult tasks could certainly be improved in future works, with the latest state-of-the-art techniques, such as with a complementary active learning process [53]. Such methodology, which contributes to interactively annotating and introducing progressively new training material, and particularly the most useful images for improving a model, could allow for a more rapid expansion of training data, which is sorely lacking at the moment.

Our study shows that historical data, with the help of AI technologies, represent an original material with which new knowledge about the vegetative phenology of tropical tree species could be built. However, the following recommendations for the community of taxonomists and biologists who collect and prepare herbarium specimens could further contribute to the facilitation of future phenological studies based on this methodological framework:Collect more specimens with growing shoots in addition to those with reproductive structures;Organize the specimens on the herbarium sheet in order to better visualize the ends of the axes, and to avoid leaf overlaps;Annotate the presence or absence of new growing shoots.

## 4. Materials and Methods

### 4.1. Assembling Herbarium Records and Manual Annotations

A specific dataset was created from a large corpus already made available to the scientific community by the Herbarium of French Guiana [54] (http://herbier-guyane.ird.fr/ - CAY - accessed on 10 May 2021) of the French national institute for sustainable development (IRD). All digitized specimens of this herbarium are accessible online (http://publish.plantnet-project.org/project/caypub_en accessed in 10 May 2021). Most of them were collected in the tropical rainforests of French Guiana, with the remaining specimens coming from Suriname, Guyana, and Brazil (Amapá and Pará states). The new dedicated dataset consists of a selection of specimens located in different ecosystems of the French Guyana territory [55]. Each specimen is annotated with taxonomic and contextual information—species, genus, family, date, locality name, and GPS—and tags about the presence of fertile organs. Initially, no information related to the presence or absence of vegetative growths was recorded for these specimens. Two experts spent more than two full-time days manually and meticulously scanning the specimens and checking for the presence or absence of new growing shoots. Analyzing and double-checking the specimen scans was a tedious task because new growing shoots are sometimes difficult to observe due to their small size or relatively discrete characteristics that only experts can notice. Finally, from an initial list of 70,563 herbarium sheets covering a large taxonomic diversity—about 3500 species, 790 genera, 130 families—1099 sheets were annotated for the presence (104 sheets) or a certified absence (995 sheets) of new growing shoots. This manual annotation gives a first estimation of the ratio of herbaria that contain growing shoots, which is about 9.46%. However, this initial estimate is probably an overestimation of the true ratio, as the two experts focused on taxonomic groups that were likely to contain growing shoots.

Moreover, for this first selection, we decided to exclude a large number of taxa from the final dataset—about 150 species, 80 genera, and 13 families—that were associated with a single specimen. Indeed, retaining too many species (and a wide taxonomic coverage) would have incurred the risk of introducing many isolated and potentially atypical vegetative growing patterns, which would have strongly affected the performances of the automatic detection methods. Furthermore, by excluding these isolated specimens, it was assured that any herbarium containing a growing shoot that would be used for the evaluation of automated approaches would have at least one training example from the same taxonomic group. In the end, the resulting dataset was composed of 408 herbarium specimens belonging to 57 species, 40 genera, and 25 families, with a total of 77 specimens showing the presence of recent growing shoots. These additional annotations of growing vegetative shoots represent the first important contribution of this paper by proposing a new annotated dataset, opening perspectives in phenological analyses based on the observation of this type of botanical structure.

During the identification of the growing shoots, we tried to distinguish two categories of shoot extension modalities: “rhythmic” or “continuous” growth. “Rhythmic growth” is manifested by an alternation of periods of rest and periods of active shoot extension or “growth flushes”. A “growth unit” is defined as the portion of an axis that develops during an uninterrupted period of extension [56]. On the other hand, “continuous” growth exhibits a relatively constant extension of phytomers, i.e., internodes associated with their leaves and axillary buds throughout the year. The temporal pattern of rhythmic growth often results in structural regularities from a morphological point of view. The limit between two growth units is usually identifiable by a series of short internodes associated with leaves reduced in size or scale leaves (i.e., cataphylls). However, in other cases, when the buds are not made of cataphylls but of leaf primordia, these “naked” buds [57] generally do not leave any morphological evidence that allow for the localization of the growth stops. In this case, it may be difficult to distinguish rhythmic from continuous growth by simple observation of the morphology. Nevertheless, when a set of leaves is elongated during a flush, they generally present a homogeneous state of maturation. On the contrary, during continuous growth, we observe a gradient of maturation (size, texture, color) from the proximal part of the elongated shoot to its distal part. In some intermediate cases, axes that elongate leaves one after the other, as in continuous growth, can enter a phase of rest constrained by the seasonality of the climate. In these cases, we assigned the category “Rhythmic” to species whose young shoots had, at their base, small internodes associated with small leaves or cataphylls in addition to young leaves with a relatively homogeneous state of maturation. On the other hand, we assigned the category “Continuous” to young shoots which did not present these characteristics. Note that this last category can include species with rhythmic growth without any morphological markers. These two growing patterns are illustrated in Figure 5. Because of the ambiguity between continuous and rhythmic patterns (due to the potential overlap of some structures with the basis of the new shoot, among others), some of the specimens initially annotated as having a recent growing shoot were finally removed from the dataset.

From the perspective of deep learning approaches, there are several ways to automatically detect recent growing shoot and eventually specify their type (“Rhythmic” or “Continuous”). In this study, we first considered a global approach that tried to detect the presence or not of growing shoots in an image without precisely indicating their locations. This first approach was then compared to local approaches that tried to more finely detect the growing shoots through bounding boxes or even masks that locate the vegetative structures at the pixel level. To evaluate the performance of such different approaches, the herbarium dataset was manually annotated at three levels of granularity related to the presence of growing shoots:**“FullImage” level.** All herbarium sheets used in this study were associated with a tag (Yes or No) indicating if the sheet had at least one recent growing shoot visible or not. No information on the location, number, or size of recent growing shoots was recorded.**“Mask” level.** In each herbarium sheet containing one or several recent growing shoots, each shoot was manually annotated and cross-validated by two co-authors using COCO Annotator (https://github.com/jsbroks/coco-annotator accessed on 10 May 2021) a web-based image annotation tool designed to efficiently label images and create training data for object detection tasks (see Figure 6). This allows for the precise capture of their full shape. It should be noted that it was not uncommon for growing shoots to be partially overlapped by other plant parts (leaves of adult sizes, branches, petioles) or sometimes by pieces of tape. In this case, only the visible part in the foreground was annotated, thus generating a mask potentially based on several disjointed polygons for a single growing shoot.**“BoundingBox” level.** Once the masks were created, the bounding boxes were directly deduced from the masks based on the min and max coordinates in x and y, respectively, for each mask. This gives us rectangles that capture both the growing shoots and the surrounding visual context such as parts of stems, leaves, textual description, and paper.

### 4.2. Preliminary Analysis of the Annotations

Before trying to predict the presence of new growing shoots in herbarium specimens, we propose to have a look at the collected annotations to gain insights into this detection task.

In Figure 7a, we superpose the growing shoots of all the scans. It can be noticed that most of them are on the upper part of the scans. Furthermore, they are more likely to be horizontally situated near the center rather than close to the edges of the sheets. This position distribution is unsurprising as, when compelled to agree with the standards, plants are often arranged in a predictable way, and if growing shoots are present on them, they will thus be found in similar areas of the sheets. This hint could be leveraged by automatic detection algorithms as it is not necessary to look at the whole scan but only at a subpart of it.

Next, we analyze the distribution of the number of growing shoots in the specimens in Figure 7b. Most scans only contain a single new growing shoot (68%), and merely a handful of them contain more than three (7.8%). However, in nearly one-third of the cases, there is more than one growing shoot in the image. While standard classification methods run on full images would see those scans as single data points, each growing shoot could be seen as a separate training sample to train from by more local approaches. Exploiting this would allow for the optimal use of the limited amount of training data available.

Finally, we study the surface occupied by each growing shoot in the image to understand the size of the objects to be detected. Figure 8a shows the distribution of the relative (average) surface occupied by each growing shoot in the scans. Most growing shoots are very small and occupy less than 1% of the pixels in the image. However, in some cases, they can cover a non-negligible part of the scan but very rarely over 10% of the total surface. Moreover, as shown in Figure 8b, there is a correlation between the number of growing shoots and their relative size. In particular, when a lot of shoots are present, they are more likely to be small. The smallness of the objects we want to detect is likely to cause issues in traditional classification approaches which are designed to detect rather big objects in the images [58].

To summarize this section, we can state that (i) the growing shoots are not randomly positioned in the scans, (ii) they can be numerous in each image, and (iii) they are usually very small relative to the image. These three observations justify looking beyond traditional image classification models, which usually look at the full image scale, and the testing of more local object detection approaches.

### 4.3. Description of Experiments

The objective of the experiments is to evaluate deep learning-based automated techniques to detect new growing shoots in herbarium specimens of tropical woody plants. More precisely, the main tasks are the following:**Detection**: What is the performance of automatic growing shoot detection in herbaria? Do we obtain performances comparable to the automatic detection of reproductive structures? With what proportions of missed growing shoots and detection errors?**Detection and classification**: Is it possible to both detect and classify different types of growing patterns (“Continuous” or “Rhythmic”) automatically?

### 4.4. Evaluated Deep Learning Architectures

Several deep learning architectures were evaluated. They can be divided into two categories: *global* and *local* approaches. The global approach considers the detection task as a visual classification task and works at the scale of a whole image without specifying which sub-parts belong or not to the different considered categories. In the context of this study, the **Detection** task is related to a two-category problem (“Without” or with “New Growing Shoot”), and to a three-category problem for the **Detection and classification** task (“Without”, “Continuous”, “Rhythmic”). The advantage of this approach is that it does not require expensive manual annotation work inside each image of the training set. A local approach, on the other hand, is specifically designed to locate objects, growing shoots in our case, with different levels of precision, such as bounding boxes or fine contour delineations, depending on the power of the method used and the manual annotation effort one is willing to make to train a deep learning model. A total of three architectures were evaluated, one using a global approach and two a local approach, all relying on the same pretrained backbone model (a ResNet50 [59]), to make fair comparisons.

**Global model (ResNet50).** The first deep learning model that was trained is the Convolutional Neural Network (CNN) ResNet50 [59]. It was pretrained on the ImageNet dataset [60] and fine-tuned on our training dataset. ResNet50 is widely used in image classification tasks and research works for its good compromise between performance, memory use, and training time. Moreover, ResNet50 is often preferred to other recent architectures for a wide range of application studies because its architecture is rather simple, and it is relatively easy to find training hyperparameters that produce good and stable results. A CNN produces as outputs a list of classification scores (*probabilities*) related to the considered categories, but without any information about the sub-parts of the image that contributed to the prediction. Pre-trained models can be easily found for most Deep Learning frameworks, particularly for PyTorch (https://pytorch.org/, accessed on 10 May 2021), which was used for our experiments. Details on this model adaptation, data augmentation strategy, and the used hyperparameters are provided in Appendix A.**Local model (Faster R-CNN).** The second model that we evaluated is based on the Faster R-CNN architecture [61], which was chosen for its demonstrated efficiency in various object detection tasks and challenges such as MS COCO [62]. A trained Faster R-CNN model produces as outputs a list of bounding boxes associated with probabilities related to the considered categories for detection. We used the Detectron2 implementation [63] itself using the PyTorch framework, based on ResNet50 as the backbone CNN and the Feature Pyramid Network [64] as the Region Proposal Network for object detection. The total number of training iterations was made based on the empirical observation of the model’s training performance. A detailed description of the hyperparameters that were used to train the model is provided in Appendix B.**Local model (Mask R-CNN).** The third model that we evaluated is based on the Mask R-CNN architecture [65], which was chosen for its ability to perform an instance segmentation task by extending the Faster R-CNN approach to a pixel-level mask prediction task. A trained Mask R-CNN model produces as outputs a list of polygon sets, each associated with probabilities related to the considered categories. As for the Faster R-CNN, we used the Detectron2 implementation [63], using by default the same backbone ResNet50 and the Feature Pyramid Networks as for Faster R-CNN. A detailed description of the hyperparameters that were used to train the model is provided in Appendix B.

### 4.5. Assessing Raw Performances of Deep Learning Models

#### 4.5.1. Training and Test Datasets

To evaluate the ability of the different models to automatically detect growing shoots, the subset of plant observations containing at least one herbarium sheet that showed growing shoots was randomly split at a ratio of 0.66, following a stratified strategy based on taxa. In this way, we reduced the risk of bias by preventing images belonging to the same plant observation from being split between the training and test sets, which would have led to overly optimistic measures of performance. On the other hand, the stratified split by taxa guaranteed that the growing shoots related to one given taxon to be detected in the test set have some training examples related to the same taxon in order to not unfairly penalize the performance. Afterwards, images without growing shoots where added to extend the training and test sets and to introduce “negative” examples that were needed to train the global model (see Table 1 for detailed statistics). For local approaches, negative images were not mandatory because this type of approach already learned to differentiate the areas of interest to be detected from the areas to be excluded within the “positive” images (i.e., images with growing shoots). However, as for the global approach, negative images were here used in the local approaches under the assumption that these images would show supplementary visual content and to train better models. For the test set, negative images allowed for the evaluation of the robustness of the approaches by measuring in particular the false positive rate.

#### 4.5.2. Tasks

To allow the comparison of our three deep learning models, a first set of of experiments were conducted to evaluate the **Detection** task, i.e., the ability to detect if an image contains or not growing shoots. Then, the same experiments were repeated on the **Detection and Classification** task, i.e., the ability to detect if an image contains a “Continuous” or a “Rhythmic” or no growing shoots.

EXP1: Detection, Global model (ResNet50)EXP2: Detection, Local models (Faster R-CNN, Mask R-CNN)EXP3: Detection and Classification, Local models (Faster R-CNN, Mask R-CNN)

#### 4.5.3. Metrics

Considering a herbarium collection, a phenological study of the vegetative growth dynamics must be able to rely on images that are deemed certain to contain growing shoots. Nevertheless, it is equally important to ensure with certainty the absence of growing shoots, especially in the case of rhythmic shoots. Indeed, too many false positives, i.e., images wrongly predicted to have growing shoots, could strongly degrade the conclusions of such a study (in particular, on the duration between stop and restart of growth). This is why we retained the main metric of the Receiver Operating Characteristic (ROC) curve. The ROC curve basically gives True Positive Rate (TPR) versus False Positive Rate (FPR) at various probability thresholds. It is typically used in binary classification, as in our experiments EXP1 and EXP2, and it can be extended to multi-label classification with one ROC curve drawn per label (“Continuous” and “Rhythmic” for the experiments EXP3). The Area Under the Receiver Operating Characteristic Curve (ROC AUC) summarizes the ROC information in one number that can be used to select the best model parameters (see Appendix B). The highest difference between the TPR and FPR is used to select the probability threshold giving the optimal cutoff point, i.e., the threshold giving the best trade-off between TPR and FPR.

## 5. Conclusions

The vegetative phenology of plants is an area where much knowledge remains to be discovered, especially for tropical floras. We still do not know the precise growth patterns for the vast majority of species, nor their adaptation behavior in the climatic, geographical, and environmental contexts. These are fundamental research questions that also have an applicative scope in the fields of silviculture and agroforestry, such as knowing the best periods of the year that favor the pruning of trees for better growth and better production of wood, or in the management of trees used for shade in coffee plantations in an agroforestry system.

We have shown in this study that herbaria represent a potentially interesting and usable source of information that can serve as material for studies on the vegetative phenology of tropical tree species. However, specimens with growing shoots are quite rare and unbalanced at the taxonomic level since the collection of specimens is often motivated by the presence of reproductive organs. We have shown that deep learning techniques can potentially help to detect at best about 87–90% of the images that truly contain growing shoots but at the cost of a false positive rate of about 28–30%. Future investigations can focus on reducing the false positive rate to ensure the prediction of the absence of growing shoots, which is a crucial piece of information, especially in the case of rhythmic shoots.

Automatic techniques can thus be used to enhance past historical data and provide original material based on herbarium specimens of great value for future works on vegetative phenology. Looking towards the future, the study also led us to propose several recommendations for the next collection of herbarium specimens in order to facilitate the implementation of automated methods, and to encourage the consideration of future phenological studies at unprecedented time and space scales that were until now deemed unthinkable by the biology community.

## Figures and Tables

**Figure 1 plants-11-00530-f001:**
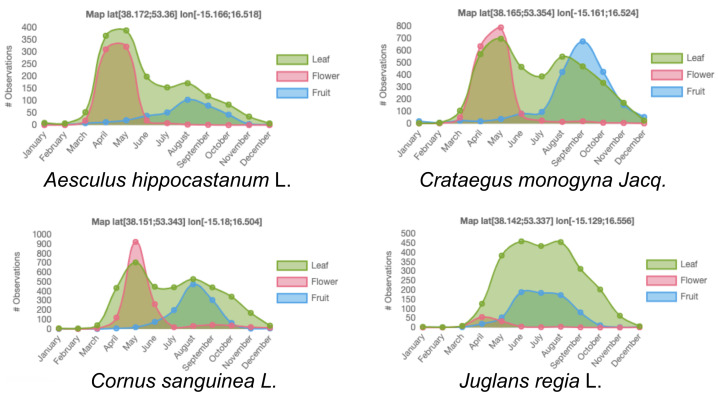
Phenological diagrams of four common tree species (*Aesculus hippocastanum* L., *Crataegus monogyna* Jacq., *Cornus sanguinea* L., *Juglans regia* L.) in western European temperate regions, based on field observations from the Pl@ntNet citizen science platform. Coordinates of the geographical areas on which these phenological diagrams are based are provided above of each of the diagrams.

**Figure 2 plants-11-00530-f002:**
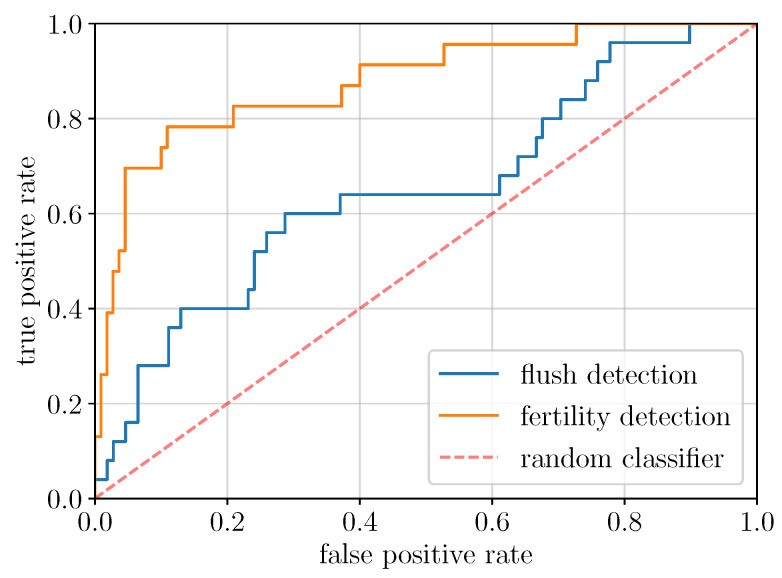
ROC curves—true positive rate versus false positive rate—of the global model for growing shoots (blue) and fertility (orange) detection. The dashed red line, representing the performance curve of a random classifier, is shown for comparison. The global model is better than the random classifier for growing shoot detection but is very far from its performance on the fertility detection task.

**Figure 3 plants-11-00530-f003:**
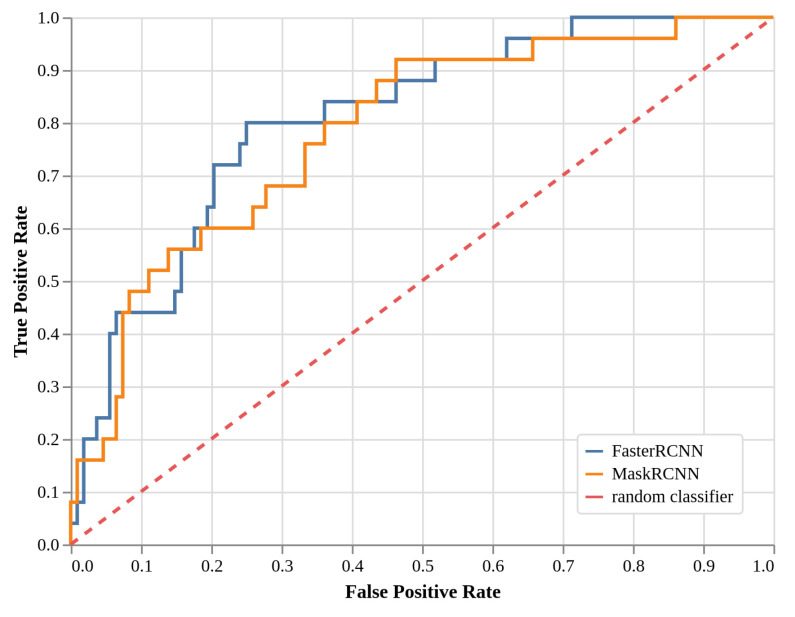
Receiver Operating Characteristic (ROC) curves—true positive rate versus false positive rate—for the Faster R-CNN (blue) and Mask R-CNN (orange) models, considering growing shoots as one class to be detected. In the dashed red line, the performance curve of a random classifier is shown for comparison. The Faster R-CNN model is better than the Mask R-CNN model, with the highest ROC AUC and a clearer optimal cut-point at TPR 0.92 and FPR 0.25. Both models are better than the global model.

**Figure 4 plants-11-00530-f004:**
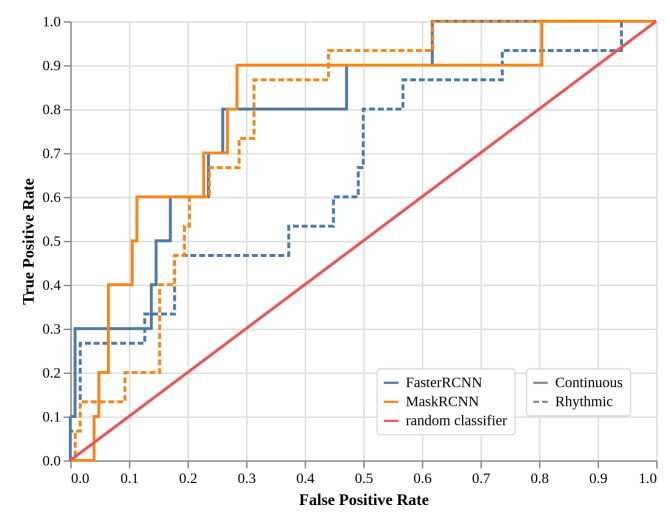
ROC curves—true positive rate versus false positive rate—for the two models considering two categories of growing shoots during the training: Faster R-CNN “Continuous” (blue), Faster R-CNN “Rhythmic” (dashed blue), Mask R-CNN “Continuous” (orange), and Mask R-CNN (dashed orange). In red, the performance curve of random classifiers is shown for comparison. The Mask R-CNN model is better than the Faster R-CNN model, especially in the “Rhythmic” category.

**Figure 5 plants-11-00530-f005:**
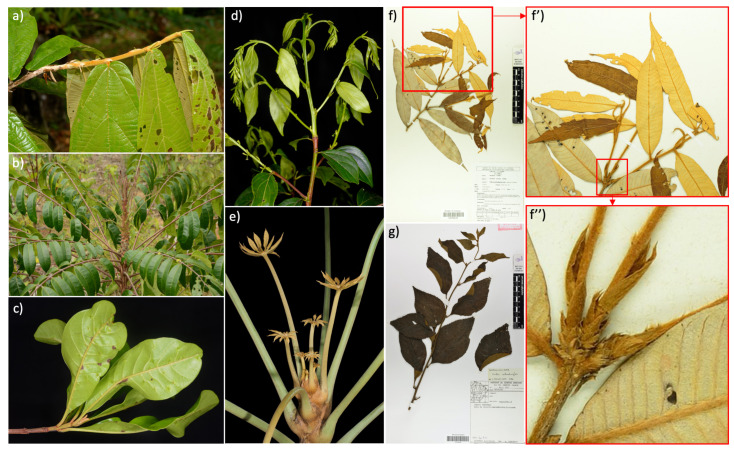
(**a**) Young shoot at the end of growth in Cupuassou (*Theobroma grandiflorum*, *Malavaceae*). Rhythmic growth is denoted by the presence of leafless internodes at the base of the growth units and by young leaves in the same state of maturation. (**b**) Growth unit at the very beginning of elongation in *Carapa procera* (*Meliaceae*). (**c**) Growth unit at the end of elongation in *Pradosia cochlearia* (*Sapotaceae*). In this case, the boundary of the growth unit is not distinguished by the presence of short internodes or cataphylls, although growth is rhythmic. (**d**) Elongating growth unit in *Goupia glabra* (*Goupiaceae*). In this case, the growth unit is branched, with the lateral branches developing at the same time as the supporting axis (immediate branching). (**e**) Gradient of size and state of maturation of the leaves from the apex to the base of the axis in *Schefflera morototoni* (*Araliaceae*). In this case, the leaves are progressively elongated one after the other, although growth may stop regularly. (**f**) Collection by D. Loubry (n°1005, CAY) of a *Parinari excelsa* (*Chrysobalanaceae*). Note the presence of a growth unit distinguished by a leaf cohort (**f′**) and the presence of cataphylls at the base associated with short internodes (**f″**). This specimen is classified as “rhythmic growth”. (**g**) M.S. collection (n°253, CAY) of a *Cordia schomburgkii* (*Boraginaceae*). On this specimen, an acropetal gradient of leaf size and maturation status is seen, with no growth unit boundary clearly visible through the presence of short internodes or cataphyll. This specimen is classified as “continuous growth”.

**Figure 6 plants-11-00530-f006:**
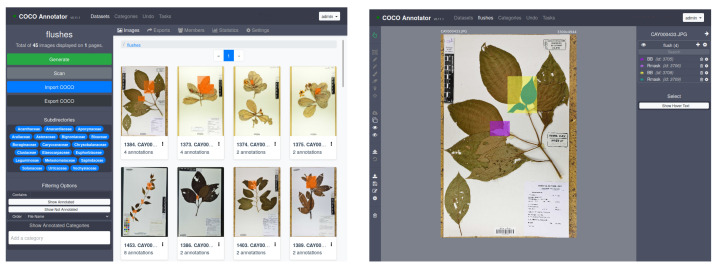
Coco annotator user interface. On the right panel, a herbarium specimen is fully annotated, with two pairs of young leaves, both with bounding boxes (yellow and purple) and manual segmentations (green and pink).

**Figure 7 plants-11-00530-f007:**
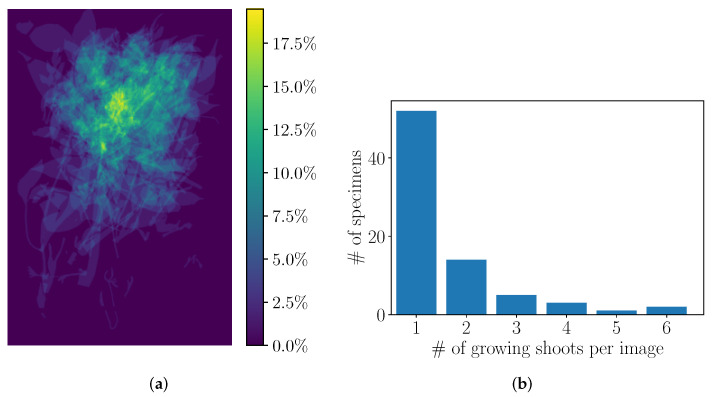
(**a**) a heatmap shows the relative distribution of the positions of the growing shoots in the herbarium specimen sheets based on the Mask level annotations. Growth shoots are not located randomly in the scans but rather in the upper part of the images. On (**b**), the distribution of the number of growing shoots per image is displayed. Most herbarium specimens contain a single shoot, and it is rare to find more than three shoots on a single sheet.

**Figure 8 plants-11-00530-f008:**
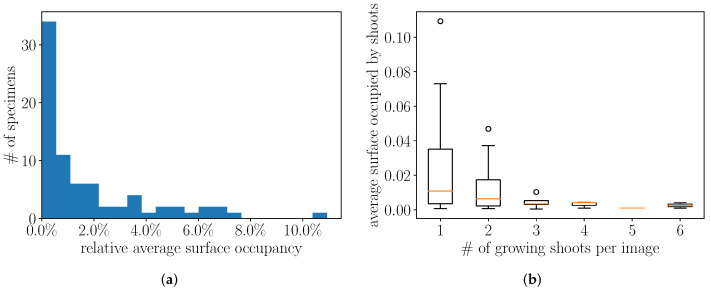
(**a**) shows the distribution of the (average) surface occupied by the growing shoots in the scan is shown. Most growth shoots are small objects within the overall image. On (**b**), this distribution is shown according to the number of growing shoots per image. The more growing shoots are present in the image, the more likely they are to be small.

**Table 1 plants-11-00530-t001:** Summary of the statistics of the dataset with the number of families (Fam.), genera (Gen.), species (Sp.), and observations (Obs.) as well as the total number of images (Total), of images with new growing shoots (w/Shoots), and of images by type of growing pattern (w/Cont. for “Continuous”, w/Rhyt. for “Rhythmic”). Note that some images have not been fully determined at the species and/or genus level.

					Images			
Set	Fam.	Gen.	Sp.	Obs.	Total	w/Shoots	w/Cont.	w/Rhyt.
**Total**	25	59	57	352	408	77	32	45
**Train**	23	49	42	235	275	52	22	30
**Test**	19	34	35	117	133	25	10	15

**Table 2 plants-11-00530-t002:** Comparison of the True Positive Rate (TPR) at three fixed False Positive Rate (FPR) values. The highest values for each FPR are in bold.

		FPR		
	Model	0.10	0.20	0.30
**TPR**	ResNet50 (global)	0.24	0.40	0.60
	Faster R-CNN (local)	0.44		**0.80**
	Mask R-CNN (local)	**0.48**	0.60	0.68

**Table 3 plants-11-00530-t003:** Comparison of the True Positive Rate (TPR) at three fixed False Positive Rate (FPR) values for the two detection task classes. The highest values for each FPR and each growing shoot type are in bold.

			FPR		
	Growing Shoot Type	Model	0.10	0.20	0.30
**TPR**	“Continuous”	Faster R-CNN	0.30	**0.60**	0.80
		Mask R-CNN	**0.40**	**0.60**	**0.90**
	“Rhythmic”	Faster R-CNN	**0.27**	0.47	0.47
		Mask R-CNN	0.20	**0.53**	**0.73**

## Data Availability

The dataset used for the experiments, the images, the metadata, and the local annotations are available in a package on Zenodo at https://zenodo.org/record/5777028 (accessed on 10 December 2021).

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
