# Peer review of "Can Artificial Intelligence Help in the Study of Vegetative Growth Patterns from Herbarium Collections? An Evaluation of the Tropical Flora of the French Guiana Forest"

_plants, 2022, doi:10.3390/plants11040530_

Round 1

Reviewer 1 Report

The article presents the application of deep learning techniques to identify vegetative structures in herbarium collections. The applied techniques are quite recent and relevant to the type of study conducted. In general the results are promising but these are only tested for a small data set so further study is required. Moreover, further effort is required to reduce the false positive rate by collecting more data as the data seem to be imbalanced.

Author Response

Thank you for your feedback. Indeed, you point out the two main problems inherent to our study:  (i) lack of test data, and, (ii) unbalanced training data.

To our knowledge, this is the first time that a study has looked at the application of deep learning techniques for studying the phenology of vegetative organs in herbaria. We tried to highlight in the paper that it was difficult to build large training and test datasets given that sample collections in herbaria are driven primarily by the presence of reproductive organs (and not growth shoots). We hope that this work can be extended in the future to more data for more extensive evaluations. To this end, in subsequent work, the models presented in the paper could be used to help detect new herbarium sheets containing vegetative shoots in various digitized herbarium collections and thus build more complete and consistent training and testing datasets. To extend this idea, we also mentioned in the discussion the possibility of using an interactive approach based on active learning which would allow to progressively reinforce the prediction model while jointly increasing the volume of annotated images.

Reviewer 2 Report

This article describes a framework for detecting and classifying vegetative structures in herbarium collections using deep learning. The extraction of such information using herbarium data is an exciting concept. The article demonstrates encouraging findings. I have the following minor comments.

1) For the sake of completeness, it is preferable to show representative phenological diagrams illustrating the diversity of tropical species as in Figure 1.

2) Caption (c) appeared twice in figure 2, while (d) is missing.

3) On page 5, lines 208-218, a bounding box annotation can be easily obtained from the mask annotation. It is unclear why a separate COCO annotation of the bounding box was conducted.

4) 237 line "This justifies the use of more advanced detection algorithms as we are looking for potentially more than one object in each image." The authors' definition of "more advanced detection algorithms" is unclear. Why is it difficult to detect multiple objects simultaneously?

5) Line 246-247, "The smallness of the objects we want to detect is likely to cause issues to traditional classification approaches which are designed to detect rather big objects in the images."   Kindly include a citation for this assertion.

6) According to page 12, table 2, faster-RCNN outperforms mask-RCNN. The annotation data for Faster-RCNN includes undesirable components such as stems, leaves, and background paper, but it performs better. Kindly explain this.

Author Response

Thank you for your positive and encouraging review, as well as your numerous suggestions, remarks and feedback.

We provide the answers to your comments in the following.

  1. We would have liked to be able to produce diagrams equivalent to figure 1 on some species of the Guyanese flora. However, we quickly run up against the lack of data to be able to produce statistically reliable curves which would make a trend in the phenological patterns. Indeed, whether via the Pl@ntNet databases, or even those aggregated via the GBIF, it is difficult to find data (photos or herbaria) gathering the 3 necessary criteria (date, geolocation and an explicit mention of the presence of flower, fruit, leaf organs, etc.) that allow to produce such curves. We hope that the work we have initiated will facilitate the extraction and aggregation of this basic information in the longer term in order to establish a better knowledge of the diversity of vegetative patterns.
  2. Thank you for pointing out this error to us, the second caption (c) was for the sub figure (d). This issue has been corrected.
  3. Thank you for highlighting this point. In that part of the manuscript, we have reported the work awkwardly in chronological order. Because this type of annotation was more time-efficient to collect, we first worked on bounding boxes to train Faster R-CNN models to estimate the order of magnitude of the performances to be expected. Then, we spent more time annotating the images to produce masks and train Mask R-CNN models. However, in the end, the boxes surrounding the masks match closely those of the initial bounding boxes annotations. The differences are negligible (up to a few pixels) and we can reasonably say that they would not have an impact on performance. We have modified this part of the paper and made it more fluid by stating that the bounding boxes have been extracted from the masks.
  4. This sentence is indeed unclear. In fact, the task is not strictly harder for those scans. However, using local detection models rather than global classification ones (taking the whole images as input) allows to optimize the use of the training data. Indeed, for those local models, each growing shoot is seen as a separate training instance. Due to the limited amount of scans with growing shoots, leveraging this point is relevant and of interest for our study. We have rephrased this part to clarify this point. 
  5. Thank you for pointing out the absence of reference to the literature. We added a citation to back this assertion. 
  6. Thanks for this suggestion, we have added a fourth additional paragraph in the discussion section. “In the case of a single general category of growing shoots, it seems that the Mask R-CNN approach encounters more difficulties to converge than the Faster R-CNN approach, probably because we force the mask prediction learning on more heterogeneous visual concepts, which are more difficult to generalize and would probably require more data to obtain better performances. Conversely, if the two concepts of growing shoots are considered as two distinct categories, the Mask R-CNN approach can more easily capture the outline of both types of growing shoots and can better converge and finally outperform the Faster R-CNN.”